# Assessment of knowledge and perception of prescribers towards rational medicine use in the Ashanti Region of Ghana

Richard Delali Agbeko Djochie[1]*, Rita Owusu-Donkor[2], Elizabeth Modupe d'Almeida[3], Francis Kwadwo Gyamfi Akwah[4‡], Emmanuel Kyeremateng[2‡], Samuel Opoku-Afriyie[5‡], Cecilia Akosua Tabiri[6‡], Francis Kyei-Frimpong[7‡], Samuel Dwomoh[2‡], Jonathan Boakye-Yiadom[8]

1 Pharmacy Department, Bekwai Municipal Hospital, Bekwai Ashanti, Ghana, 2 Ashanti Regional Health Directorate, Kumasi, Ghana, 3 Pharmacy Department, Asonomaso Government Hospital, Asonomaso, Ghana, 4 Atwima Nwabiagya Municipal Health Directorate, Nkawie, Ghana, 5 Pharmacy Department, Effiduase Government Hospital, Effiduase, Ghana, 6 Pharmacy Department, Manhyia Government Hospital, Kumasi, Ghana, 7 Pharmacy Department, Ashanti Regional Hospital, Kumasi, Ghana, 8 Komfo Anokye Teaching Hospital, Kumasi, Ghana

☯ These authors contributed equally to this work.
‡ FKGA, EK, SOA, CAT, FKF and SD also contributed equally to this work.
* richarddjochie@gmail.com

## Abstract

### Background

Prescribers must possess extensive knowledge and maintain a positive attitude towards the rational use of medicines to achieve desirable treatment outcomes and effectively prevent treatment failures, increased costs, drug toxicities, and interactions. The objective of this study was to evaluate prescribers' understanding and perception concerning the rational use of medicines in public hospitals. Additionally, the study aimed to identify the factors that influence rational prescribing practices.

### Methods

A structured data instrument was developed to collect demographic data and evaluate participants' knowledge and perception of rational medicine use, in line with the study objectives. Chi-squared statistics and Fisher's exact test were utilized to identify factors associated with good knowledge and perception among participants. Logistic regression was then employed to assess the strength of the associations, with odd ratios reported at a significant level of 0.05.

### Results

Out of 192 participants, 85.4% held a positive view of rational medicine use, stressing patient safety and recognizing risks like antimicrobial resistance and polypharmacy. Perception was influenced by factors such as prescriber profession, access to references, and drug bulletin updates. Additionally, 65.6% demonstrated good knowledge of rational medicine use, which was notably influenced by factors like using standard prescribing guidelines,

**Data Availability Statement:** All data sets associated with this manuscript have been added as a Supporting information.

**Funding:** The author(s) received no specific funding for this work.

**Competing interests:** The authors have declared that no competing interests exist.

having a functional Drug and Therapeutics Committee, prescriber profession, and the frequency of drug bulletin updates.

## Conclusion

The study emphasizes the critical need to address knowledge gaps among healthcare professionals, especially nurses and other prescribers, to ensure the safe and effective use of medications. It highlights the positive influence of utilizing preferred prescribing references and the existence of functional Drug and Therapeutics Committees in hospitals on knowledge levels. However, the unexpected findings regarding the limited impact of frequent updates of drug bulletins require further investigation.

## Introduction

Rational prescribing practices have been recognized as crucial to enhancing healthcare outcomes and decreasing healthcare expenses [1]. To achieve this goal, it is imperative for prescribers to possess extensive knowledge, along with a positive attitude and perception towards rational medication use [2, 3]. It is widely acknowledged that the excessive or improper utilization of drugs can lead to treatment failures, escalated treatment costs, drug toxicities, and drug interactions [1, 4, 5]. Although prescribing is commonly viewed as a routine task, it is a complex procedure requiring healthcare providers to possess sufficient knowledge and adhere to sound therapeutic principles. Effective communication skills and a proper understanding of risks and uncertainties are also essential [6].

The prescribing process often begins by establishing the desired therapeutic goals, such as reducing fever, eliminating an infection, or providing contraception. The goals may be influenced by patient expectations and preferences. Once the goals are determined, a suitable treatment is chosen, which can be challenging due to the various available options [7]. Ideally, the final selection of medication should be made after conducting a comprehensive benefit—risk analysis, considering both the medical factors and the patient's circumstances, including availability and cost [6, 8, 9]. Patient-related factors, such as physiological conditions (e.g., allergy, liver impairment), susceptibility to adverse effects, and concurrent drug therapy, may influence the medicine selection process by potentially leading to drug interactions [9, 10]. Additionally, drug-related factors, such as safety and efficacy evidence, as well as pharmacokinetic and pharmacodynamic properties, may also play a role in the selection process. For example, a medication with a once-daily dosing schedule might be preferred over one with multiple doses to enhance patient compliance, especially in the elderly who are likely to forget and therefore miss some doses [9, 11].

Consequently, the prescriber plays a pivotal role in implementing policies for rational medicine use, ensuring patient safety. However, numerous studies have identified gaps in prescriber knowledge and perceptions regarding rational prescribing practices in low- and middle-income countries (LMICs) [12–16]. For instance, in research carried out in Pakistan [17], it was reported that more than 60% of general practitioners (GPs) depend on pharmaceutical company representatives to receive updates on antihypertensive medications. Additionally, over 40% of GPs inappropriately prescribe sedatives to elderly patients [17]. Moreover, 23% of GPs mistakenly cease treatment once they have successfully achieved blood pressure control [17]. Likewise, another investigation carried out in healthcare facilities in rural Burkina Faso revealed that merely 50% of the prescribed doses of antimalarial medications were in

accordance with the recommendations, while antibiotics were prescribed at approximately 200% higher than the recommended doses [18]. Consequently, this led to treatment failures in the case of antimalarials and undesired effects in the case of antibiotics [18].

Prescribing is a skill that is honed through years of practical experience, as it is rarely taught comprehensively in schools. In fact, junior doctors often lack the confidence to prescribe medications [19, 20], as it is a complex process that requires the ability to consider individual and regulatory factors to avoid suboptimal prescribing [21]. There is ample evidence demonstrating the widespread occurrence of irrational prescribing practices worldwide, particularly in developing countries [22–24]. These practices encompass polypharmacy, inappropriate use of antibiotics and injections, prescribing expensive branded medications when unnecessary, and more. The consequences of such irrational prescribing practices are numerous, including patients failing to adhere to treatment due to adverse effects resulting from drug interactions, and an increase in hospitalizations due to these adverse effects [25–27]. As a result of irrational prescribing, patients lose confidence in the healthcare system and may turn to unorthodox treatments when their quality of life is affected [28].

Rational medicine usage in the Ashanti Region of Ghana has witnessed notable advancements in the past five years, particularly concerning the prescription of generic medications from the essential medicines list and the promotion of safe injection practices [29]. However, healthcare authorities and policymakers continue to grapple with significant concerns surrounding inappropriate antibiotic usage and polypharmacy, which demand urgent attention [29]. Possessing a good knowledge and perception of rational prescribing equips physicians with the skills and understanding necessary to make informed decisions about medication use [30]. It enhances patient safety, improves treatment outcomes, optimizes resource utilization, supports antimicrobial stewardship, promotes patient-centred care, facilitates adherence to guidelines, and underscores professional competence [1, 31–33].

There is a scarcity of research regarding the knowledge and perception of rational prescribing among physicians in Ghana. However, the prescribing of medications in Ghanaian hospitals involves various categories of healthcare professionals, including doctors, physician assistants, nurses, mental health nurses, and disease control officers, among others. These diverse prescriber groups may possess varying knowledge bases and perceptions concerning rational prescribing, and their practices can have an impact on patient safety and the emergence of antimicrobial resistance. Given the extent of inappropriate antibiotic prescriptions, the prevalence of polypharmacy, and the rising expenses associated with the preference for branded medications in the region [29], it becomes imperative for health authorities and policymakers to gain a comprehensive understanding of the knowledge level of these prescribers on rational use of medicines (RUM). Consequently, the objective of this study was to assess the knowledge and perception of prescribers in public hospitals in the Ashanti Region regarding the RUM, as well as to explore the factors associated with their understanding and practices. The findings from this study will inform what measures should be implemented to improve rational prescribing in the region to ensure patient drug safety.

## Methods

### Study design and sampling

This study employed a cross-sectional design and included prescribers from public primary and secondary hospitals in the region. Medication prescribing in Ghanaian hospitals involves a diverse range of healthcare professionals, comprising doctors, physician assistants, nurses, midwives, mental health nurses, community health nurses and disease control officers. Prescribers who are general nurses and midwives were categorized as "nurses" and community

health nurses, mental health nurses and disease control officers were classified as "others". While there exists a register of doctors and physician assistants, the same cannot be said for other types of prescribers. Consequently, it was challenging to determine the exact number of prescribers in public hospitals within the region. Therefore, all prescribers in the 25 public hospitals in the region were approached for participation and only those who willingly agreed to take part in the study and provided informed consent were included as participants.

## Data collection and analysis

A structured questionnaire with 34 items was developed in line with the study objectives to collect demographic information from participants and evaluate their knowledge and perception of rational prescribing. The data instrument used in this study is available in the supporting information section as S1 Tool. The reliability and validity of the questionnaire were assessed to ensure the robustness of the study findings by pretesting the data collection instrument among prescribers in one district hospital. Data collection for the main study was conducted by research assistants personally delivering the questionnaires to the prescribers in their hospitals. Participants were allowed the entire day to complete the questionnaire. The completed questionnaires were then retrieved at the end of the day. Participant recruitment and data collection took place between August 10th and September 10th, 2023.

To evaluate participants' knowledge, a selection of eleven questionnaire items (Q14—Q24) was utilized. These items encompassed understanding the distinction between generic and proprietary names, familiarity with guidelines for safe injection prescribing and nonpolypharmacy, and knowledge regarding the rational usage of antibiotics. The RUM knowledge score was calculated by summing the correct answers for questions 14 to 24, with each correct answer earning one point. The total points accrued were then converted into percentages by dividing by 11 (the maximum possible score) and multiplying by 100 and rounded to one decimal place using standard rounding rules. Respondents who achieved a score of 70% or higher were classified as having a good RUM knowledge while any score below 70% indicated poor knowledge.

The participant's perception of RUM was assessed using a Likert scale comprising six items (items 25–30). These questions aimed to gauge participants' attitudes and beliefs regarding various aspects of RUM, such as appropriateness of prescribing (item 25), adherence to guidelines (item 26), patient-centred care (items 27 and 29), and avoidance of polypharmacy (items 28 and 30). Responses ranged from "Strongly Disagree" to "Strongly Agree," coded as 1–5. A higher score indicated a more positive perception. A composite perception score was calculated by summing the responses to all six items. Scores of 24–30 were classified as good perception, 18–23 as neutral perception, and any score below 18 as poor perception.

Functional Drug and Therapeutic Committee (DTC) status was determined based on participants who responded affirmatively to at least three of the following questions: awareness of a DTC operating in their hospital (item 31), knowledge of a recent RUM survey conducted within the past six months in their hospital (item 32), familiarity with the findings of the RUM survey in their hospital (item 33), and participation in a RUM training or refresher course held within their hospital (item 34).

The collected data underwent cleaning using Microsoft Excel 2016 and was then imported into Stata version 17 for analysis. Microsoft Word 2016 was utilized to create charts and tables. Categorical data were reported in frequencies and percentages, with Chi-squared or Fisher's exact test used for comparisons. Continuous data were presented as mean (SD) or median (interquartile range). Binary logistic regression (odds ratios) compared knowledge types among participant factors such as age, gender, type of hospital, and prescriber category.

Ordinal logistic regression analyzed participants' odds of transitioning from neutral to good perception. Statistical significance was set at p < 0.05.

The dataset used to analyse the knowledge and perception of prescribers towards RUM is provided as a supplementary material in the supporting information section (S1 Dataset).

### Ethical consideration

Prescribers who voluntarily agreed to participate in this study provided informed consent after a comprehensive explanation of the study's objectives. To safeguard participant confidentiality, all identifying information was deliberately excluded during the data collection process. Prescribers were granted ample privacy and flexibility to respond to the questionnaire, with the entire day available for submission. This study protocol received ethical approval from the ethics committee at Kwame Nkrumah University of Science and Technology and was issued a certificate bearing the number CHRPE/AP/706/23 on 8th August 2023.

## Results

### Sociodemographic characteristics of participants

Out of the 215 questionnaires that were distributed, 192 were completed and returned, resulting in a response rate of 89.3%. The participant demographics are presented in Table 1, revealing that the majority of participants were male (53.1%), married (54.8%), and identified as physician assistants (29.2%). Additionally, 90.6% of the participants worked in primary-level hospitals. The mean age of the participants was 34.4 years (±7.7). Half of the respondents had been working as prescribers for 3 years or less, while 32.1% had been in their current hospital positions for over six years.

### Participants' perception of rational use of medicines

The majority of participants demonstrated a good perception of rational medicine use, with 85.4% (n = 164) having a positive perception. Most of them prioritize patient safety over simply curing diseases, with 82.2% agreeing. Additionally, 80.7% agree that irrational prescribing contributes to antimicrobial resistance. Most respondents (81.7%) believe injections aren't inherently more effective than other forms of medication. Concerns about polypharmacy's risk of drug interactions are shared by 75.0% of respondents. Furthermore, 64.5% agree that irrational prescribing can lead to hospitalizations, while 86.9% support reserving certain medicines for specialist prescribing. Factors significantly associated with perception of rational medicine use included prescriber profession (p = 0.011), availability of reference sources (p = 0.026), frequency of drug bulletin updates (p = 0.007), and the use of Medscape as a reference (p = 0.046) (Table 2). These associations were further analyzed using ordinal regression.

Prescribers with access to reference sources were nearly three times more likely to transition from neutral to good perception (OR = 2.9; 95% CI: 1.3–6.6; p = 0.0095) (Table 3). However, compared to doctors, nurses (OR = 0.08, 95% CI: 0.01–0.61; p = 0.018), medical interns (OR = 0.04; 95% CI: 0.003–0.43, p = 0.008), and other prescribers, such as mental health nurses, disease control officers, and community health nurses (OR = 0.57; 95% CI: 0.01–0.56; p = 0.014), were less likely to improve from neutral to good perception.

Furthermore, prescribers whose drug bulletins were updated annually had significantly lower odds of transitioning from neutral to good perception compared to those whose bulletins were never updated (OR = 0.21, 95% CI: 0.07–0.62; p = 0.005).

**Table 1. Socio-demographic characteristics of study participants.**

| Characteristic | Number of respondents *n* (%) |
|---|---|
| **Age** | |
| 20–25 | 21 (11.9) |
| 26–35 | 90 (50.8) |
| 36–45 | 53 (29.9) |
| 46–55 | 13 (7.4) |
| **Gender** | |
| Male | 102 (53.1) |
| Female | 90 (46.9) |
| **Marital status** | |
| Married | 107 (55.7) |
| Single | 83 (43.3) |
| Widowed | 2 (1.0) |
| **Level of hospital of practice** | |
| Primary | 174 (90.6) |
| Secondary | 18 (9.4) |
| **Category of prescriber** | |
| Medical Officer | 47 (24.5) |
| Medical Intern | 7 (3.7) |
| Physician Assistant | 56 (29.2) |
| Physician Assistant Intern | 30 (15.6) |
| Nurse Prescriber | 36 (18.8) |
| Others | 16 (8.2) |
| **Employment status** | |
| Permanent | 146 (76.6) |
| Temporal (locum) | 8 (4.2) |
| Trainee | 35 (19.2) |
| **Years of practice** | |
| ≤3 years | 96 (50.0) |
| >3–6 years | 34 (17.7) |
| >6–10 years | 37 (19.3) |
| >10 years | 25 (13.0) |

## Participants' knowledge of rational use of medicines

The majority of participants (65.6%, n = 126) demonstrated a strong understanding of RUM. Notably, individuals working in primary hospitals exhibited significantly better knowledge compared to those in secondary (referral) hospitals (p = 0.047). Furthermore, participants who relied on the Standard Treatment Guideline (STG) and the British National Formulary (BNF) as their prescribing references showed superior knowledge of rational prescription practices, in contrast to those who used Medscape and the institutional drug bulletin (p = 0.001, p = 0.001, p = 0.057, and p = 0.174 respectively). The professional category of the prescriber (p = 0.001) and the frequency of updates of the hospital drug bulletin (p = 0.001) were also identified as significant factors influencing RUM knowledge. However, factors such as age, gender, employment status, and years of experience as a prescriber did not show significant relationships with prescriber knowledge.

In binary logistic regression analysis as depicted in Table 3, participants who utilized STG and BNF as prescription references had approximately three times the odds of possessing good

**Table 2. Comparison of participant characteristics with knowledge and perception of rational medicine use.**

| Characteristic | RUM Knowledge | | | RUM Perception | | | |
|---|---|---|---|---|---|---|---|
| | Good *n* (%) | Poor *n* (%) | *p-value | Good *n* (%) | Neutral *n* (%) | Poor *n* (%) | δp-value |
| **Drug bulletin as reference** | | | 0.174 | | | | 0.169 |
| Yes | 99 (68.3) | 46 (31.7) | | 125 (86.5) | 10 (6.9) | 10 (6.9) | |
| No | 27 (57.5) | 20 (42.6) | | 39 (83.0) | 7 (14.9) | 1 (2.1) | |
| **STG as reference** | | | 0.001 | | | | 0.315 |
| Yes | 100 (72.5) | 38 (27.5) | | 121 (87.7) | 10 (7.2) | 7 (5.1) | |
| No | 26 (48.2) | 28 (51.9) | | 43 (79.6) | 7 (13.0) | 4 (7.4) | |
| **BNF as reference** | | | 0.001 | | | | 0.609 |
| Yes | 67 (77.9) | 19 (22.1) | | 74 (86.0) | 6 (7.0) | 6 (7.0) | |
| No | 59 (55.7) | 47 (44.3) | | 90 (84.9) | 11 (10.4) | 5 (4.7) | |
| **Medscape as reference** | | | 0.057 | | | | 0.046 |
| Yes | 58 (73.4) | 21 (26.6) | | 66 (83.5) | 11 (13.9) | 2 (2.5) | |
| No | 68 (60.2) | 45 (39.8) | | 98 (86.7) | 6 (5.3) | 9 (8.0) | |
| **Reference source accessible** | | | 0.323 | | | | 0.026 |
| Yes | 89 (67.9) | 42 (32.1) | | 118 (90.1) | 8 (6.1) | 5 (3.8) | |
| No | 37 (60.7) | 24 (39.3) | | 46 (75.4) | 9 (14.8) | 6 (9.8) | |
| **Drug bulletin updates** | | | 0.001 | | | | 0.007 |
| Never | 35 (64.8) | 19 (35.2) | | 48 (88.9) | 5 (9.3) | 1 (1.9) | |
| Quarterly | 45 (69.2) | 20 (30.8) | | 60 (92.3) | 2 (3.1) | 3 (4.6) | |
| Monthly | 34 (82.9) | 7 (17.1) | | 36 (87.8) | 3 (7.3) | 2 (4.9) | |
| Annually | 12 (37.5) | 20 (62.5) | | 20 (62.5) | 7 (21.9) | 5 (15.6) | |
| **Age (years)** | | | 0.934 | | | | 0.456 |
| 20–25 | 13 (61.9) | 8 (38.1) | | 16 (76.2) | 4 (19.0) | 1 (4.8) | |
| 26–35 | 62 (64.6) | 34 (35.4) | | 83 (86.5) | 7 (7.3) | 6 (6.3) | |
| 36–45 | 39 (67.2) | 19 (32.8) | | 52 (89.7) | 3 (5.2) | 3 (5.2) | |
| 45–55 | 12 (70.6) | 5 (29.4) | | 13 (76.5) | 3 (17.6) | 1 (5.9) | |
| **Level of Hospital** | | | 0.047 | | | | 0.284 |
| Primary | 118 (67.8) | 56 (32.2) | | 149 (85.6) | 14 (8.0) | 11 (6.3) | |
| Secondary | 8 (44.4) | 10 (55.6) | | 15 (83.3) | 3 (16.7) | 0 (0.0) | |
| **Type of prescriber** | | | 0.001 | | | | 0.011 |
| Doctor | 37 (78.7) | 10 (21.3) | | 46 (97.9) | 1 (2.1) | 0 (0.0) | |
| Medical intern | 3 (42.9) | 4 (57.1) | | 5 (62.5) | 3 (37.5) | 0 (0.0) | |
| Physician assistant | 41 (73.2) | 15 (26.8) | | 48 (85.7) | 4 (7.1) | 4 (7.1) | |
| Physician assistant intern | 23 (76.7) | 7 (23.3) | | 25 (86.2) | 3 (10.3) | 1 (3.4) | |
| Nurse | 17 (47.2) | 19 (52.8) | | 28 (77.8) | 5 (13.9) | 3 (8.3) | |
| Others | 5 (31.3) | 11 (68.7) | | 12 (75.0) | 1 (6.3) | 3 (18.8) | |
| **Years of experience** | | | 0.167 | | | | 0.172 |
| ≤3 years | 63 (65.6) | 33 (34.4) | | 75 (78.1) | 11 (11.5) | 10 (10.4) | |
| >3–6 years | 20 (58.8) | 14 (41.2) | | 32 (94.1) | 2 (5.9) | 0 (0.0) | |
| >6–10 years | 22 (59.5) | 15 (40.5) | | 34 (91.9) | 2 (5.4) | 1 (2.7) | |
| >10 years | 21 (84.0) | 4 (16.0) | | 23 (92.0) | 2 (8.0) | 0 (0.0) | |
| **State of DTC** | | | 0.004 | | | | 0.394 |
| Functional | 39 (83.0) | 8 (17.0) | | 43 (91.5) | 2 (4.3) | 2 (4.3) | |
| Nonfunctional | 86 (59.7) | 58 (40.3) | | 121 (83.4) | 15 (10.3) | 9 (6.2) | |

RUM = rational use of medicine; DTC = drugs and therapeutic committee; STG = standard treatment guidelines; BNF = British National Formulary

*p-value was determined using the chi-squared statistic.

δp-value was determined using Fisher's exact test

**Table 3. Factors influencing the odds of having good knowledge and transitioning from neutral to good perception of rational medicine use.**

| Characteristics | Knowledge | | | Perception | | |
|---|---|---|---|---|---|---|
| | OR | 95% CI | *p*-value | OR | 95% CI | [δ]*p*-value |
| **Using STG as a reference** | | | | | | - |
| No | 1 | | | | | |
| Yes | 2.83 | 1.48–5.44 | **0.002** | -* | - | |
| **Using BNF as a reference** | | | | | | - |
| No | 1 | | | | | |
| Yes | 2.81 | 1.48–5.31 | **0.001** | - | - | |
| **Frequency of drug bulletin update** | | | | | | |
| Never updated | 1 | | | | | |
| Monthly | 2.64 | 0.98–7.07 | 0.054 | 0.87 | 0.25–3.08 | 0.834 |
| Quarterly | 1.22 | 0.57–2.63 | 0.510 | 1.43 | 0.41–4.98 | 0.570 |
| Annually | 0.33 | 0.13–0.81 | **0.015** | 0.21 | 0.07–0.62 | **0.005** |
| **Prescribing reference source accessible** | | | | | | **0.009** |
| No | - | - | - | 1 | | |
| Yes | | | | 2.94 | 1.30–6.64 | |
| **Functionality of DTC** | | | | | | - |
| Nonfunctional | 1 | | | - | - | |
| Functional | 3.28 | 1.43–7.54 | **0.005** | | | |
| **Level of hospital** | | | | | | - |
| Secondary | 1 | | | - | | |
| Primary | 2.63 | 0.99–7.04 | 0.053 | - | | |
| **Category of prescribers** | | | | | | |
| Doctor | 1 | | | | | |
| Medical Intern | 0.20 | 0.04–1.06 | 0.058 | 0.04 | 0.003–0.43 | **0.008** |
| Physician Assistant | 0.74 | 0.30–1.85 | 0.517 | 0.13 | 0.02–1.05 | 0.056 |
| Physician Assistant Intern | 0.89 | 0.30–2.66 | 0.832 | 0.14 | 0.02–1.34 | 0.089 |
| Nurse Prescriber | 0.24 | 0.09–0.64 | **0.004** | 0.08 | 0.01–0.64 | **0.018** |
| Others | 0.12 | 0.03–0.44 | **0.001** | 0.06 | 0.01–0.56 | **0.014** |
| **Using Medscape as a reference** | | | | | | 0.673 |
| No | - | - | - | 1 | | |
| Yes | | | | 0.84 | 0.38–1.88 | |

OR = odds ratio; CI = confidence interval; DTC = drugs and therapeutic committee; BNF = British National Formulary;

*Not statistically significant at the bivariate level. Boldface entries indicate statistically significant variables. "Others" include mental health nurses, disease control officers and community health nurses.

Binary logistic regression analysis.

[δ]Ordinal logistic regression analysis.

prescription knowledge (OR = 2.83, 95%CI: 1.47–5.44; p = 0.002) and (OR = 2.81, 95%CI: 1.48–5.31; p = 0.001), respectively. Conversely, nurse prescribers (OR = 0.24; 95% CI: 0.09–0.64; p = 0.004), along with other healthcare professionals such as community health nurses, disease control officers, and mental health nurses (OR = 0.12; 95% CI: 0.03–0.44; p = 0.001), were less likely to demonstrate good RUM knowledge compared to doctors. Additionally, participants working in hospitals with functional DTCs were more likely to possess good prescribing knowledge (OR = 3.29; 95%CI: 1.43–7.54; p = 0.002) compared to those who did not have access to such resources.

## Discussion

Overall, most participants demonstrated adequate knowledge and a positive perception of the concept of RUM. Specifically, 65.6% exhibited good knowledge in this area, while 85.4% held a positive perception. These figures surpass the rates found among postgraduate medical students in India, which were 61% for good knowledge and 51% for a positive perception [34]. Although the majority of participants in the present study possessed sound knowledge, it is essential for health policymakers to be concerned about those with insufficient knowledge (34.4%) because widespread adherence to RUM across all levels of healthcare is crucial for ensuring patient safety.

The profession of the prescriber plays a role in determining their level of understanding regarding RUM. Although there was no significant difference in knowledge between doctors and physician assistants, nurses and other prescribers exhibited lower odds of having a good knowledge compared to doctors. Additionally, a significantly smaller proportion of nurses, other prescribers and medical interns demonstrated a positive perception towards RUM and were less likely to move from neutral to good perception in comparison to doctors. This finding is consistent with prior research conducted among Ghanaian prescribers, specifically assessing knowledge of antimicrobial resistance, which revealed that doctors exhibited superior knowledge compared to Community Health Officers [35]. Similarly, a study conducted in Pakistan evaluating the knowledge, attitude, and practice of rational antibiotic use among health workers found that doctors attained higher knowledge scores than nurses [36]. These studies corroborate our findings and suggest a trend wherein doctors tend to demonstrate better RUM knowledge levels compared to nurses in similar contexts.

The discovery of RUM knowledge disparities among healthcare professionals, particularly concerning nurses and disease control officers raises significant concerns for patient safety within the health system. As lower-level health facilities rely on nurse prescribers, the potential for prescriptions falling short of required standards increases the risk of drug-related problems for patients [27]. Further research is imperative to delve into the underlying reasons behind these knowledge gaps among doctors and other healthcare professionals to address this concern. Crucially, it is paramount to ensure that nurse prescribers receive adequate training and mentoring to meet the requisite standards before independently prescribing medications [12, 30]. By investing in their professional development, healthcare systems can better equip these professionals to deliver safe and effective care, thereby safeguarding patient well-being and promoting optimal health outcomes. Furthermore, medical school curricula must enhance the training on RUM to equip newly trained doctors with a comprehensive understanding of this concept. This will ensure that they are well-prepared and up-to-date in their knowledge and application of RUM principles.

Participants who use STG and the BNF as their prescribing references exhibited better knowledge of rational prescribing, although their perception was not significantly affected. Those who rely on STG and BNF have more than twice the odds of good knowledge compared to those who do not use them. Additionally, having access to a prescriber's preferred reference also increases their odds of transitioning from neutral to good perception by almost three times. These reference sources contain guidelines with evidence-based recommendations, making healthcare providers who regularly consult them more likely to possess knowledge in RUM [3, 37]. This finding aligns with a study in Saudi Arabia [3] which reported that access to the right prescription references improved prescriber knowledge and practice of RUM. The findings on prescription references, however, contradict a study conducted in the Netherlands, where authors reported that the source of prescription reference had no impact on rational prescribing [38]. However, the Netherlands study was conducted over forty years ago when

internet access was not in existence to make reference sources freely available to prescribers. Therefore, this new finding in the current study may be attributed to the widely available and free prescribing reference sources that influence prescriber knowledge.

Furthermore, participants who work in hospitals that had access to institutional drug bulletins updated annually demonstrated lower knowledge levels and were less likely to transition from neutral to good perception compared to those with prescribers whose bulletins are never updated. This finding contradicts prevailing literature, which suggests that regular updates of the drug bulletin enhance prescriber knowledge for rational medicine use [12, 39]. Despite the bulletin's role in disseminating crucial information on services, changes, new inclusions, and updates relevant to prescribing practices, including new evidence and clinical guidelines, the anticipated positive effect on prescriber knowledge and perception was not observed in our study. Further research is needed to understand the underlying factors contributing to this discrepancy, considering the potential influences such as variations in institutional practices and the quality of information in the bulletins.

The presence of a functional DTC in a participant's hospital was found to significantly increase the odds of having good knowledge of RUM by more than three times. This finding aligns with other studies that reported improvements in rational prescribing and a reduction in medication errors when a hospital has a functional DTC [5, 12, 40, 41]. Therefore, it is crucial to empower hospitals to establish and adequately resource DTCs to effectively implement the RUM agenda and ensure patient safety.

## Study limitations

The cross-sectional design of our study limits our ability to establish causal relationships, and the specific demographics of our sample may restrict the generalizability of our findings. Additionally, self-reported data introduce potential response bias and may not entirely reflect actual behaviours. Despite these limitations, our study provides valuable insights into the field of Rational Use of Medicines research.

## Conclusion

The study revealed that most participants have a good knowledge and positive perception of RUM. However, nurses, community health nurses, mental health nurses, and disease control officers exhibited lower knowledge levels compared to doctors, indicating a need for targeted training programs within these groups. Additionally, the study found that relying on prescribing references was associated with higher levels of knowledge, and having access to preferred references improved the likelihood of having a positive perception. Moreover, the presence of functional DTCs in hospitals significantly influenced knowledge levels, emphasizing the importance of supporting hospitals in establishing and resourcing DTCs for safe medication use. Surprisingly, frequent updates of drug bulletins did not improve knowledge or perception, with prescribers whose bulletins were never updated showing better odds of having good knowledge and perception. Further research is needed to understand this discrepancy and its implications.

## Supporting information

**S1 Tool. Structured questionnaire used for data collection.**
(DOCX)

**S1 Dataset. Dataset on prescriber knowledge and perception towards RUM.**
(XLSX)

## Acknowledgments

The authors would like to express their gratitude to the Regional Director of Health, the management teams of the hospitals involved in the study, and the prescribers for their valuable and willing participation in the research.

## Author Contributions

**Conceptualization:** Richard Delali Agbeko Djochie, Rita Owusu-Donkor, Elizabeth Modupe d'Almeida, Francis Kwadwo Gyamfi Akwah, Emmanuel Kyeremateng, Samuel Opoku-Afriyie, Cecilia Akosua Tabiri, Francis Kyei-Frimpong.

**Data curation:** Richard Delali Agbeko Djochie, Elizabeth Modupe d'Almeida, Francis Kwadwo Gyamfi Akwah, Emmanuel Kyeremateng, Samuel Opoku-Afriyie, Cecilia Akosua Tabiri, Francis Kyei-Frimpong, Samuel Dwomoh.

**Formal analysis:** Richard Delali Agbeko Djochie, Elizabeth Modupe d'Almeida, Samuel Opoku-Afriyie, Cecilia Akosua Tabiri, Francis Kyei-Frimpong, Samuel Dwomoh, Jonathan Boakye-Yiadom.

**Investigation:** Richard Delali Agbeko Djochie, Elizabeth Modupe d'Almeida, Emmanuel Kyeremateng, Cecilia Akosua Tabiri.

**Methodology:** Richard Delali Agbeko Djochie, Rita Owusu-Donkor, Elizabeth Modupe d'Almeida, Francis Kwadwo Gyamfi Akwah, Francis Kyei-Frimpong, Jonathan Boakye-Yiadom.

**Project administration:** Richard Delali Agbeko Djochie, Rita Owusu-Donkor.

**Visualization:** Samuel Dwomoh, Jonathan Boakye-Yiadom.

**Writing – original draft:** Richard Delali Agbeko Djochie, Elizabeth Modupe d'Almeida, Jonathan Boakye-Yiadom.

**Writing – review & editing:** Richard Delali Agbeko Djochie, Rita Owusu-Donkor, Elizabeth Modupe d'Almeida, Francis Kwadwo Gyamfi Akwah, Emmanuel Kyeremateng, Samuel Opoku-Afriyie, Cecilia Akosua Tabiri, Francis Kyei-Frimpong.

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
