## [Decision Letter · Decision Letter 0]

8 Apr 2024

PONE-D-23-33862Assessment of knowledge and perception of prescribers towards rational medicine use in the Ashanti Region of GhanaPLOS ONE

Dear Dr. Djochie,

Thank you for submitting your manuscript to PLOS ONE. After careful consideration, we feel that it has merit but does not fully meet PLOS ONE’s publication criteria as it currently stands. Therefore, we invite you to submit a revised version of the manuscript that addresses the points raised during the review process. Please submit your revised manuscript by May 23 2024 11:59PM. If you will need more time than this to complete your revisions, please reply to this message or contact the journal office at plosone@plos.org. Please include the following items when submitting your revised manuscript:A rebuttal letter that responds to each point raised by the academic editor and reviewer(s). You should upload this letter as a separate file labeled 'Response to Reviewers'.A marked-up copy of your manuscript that highlights changes made to the original version. You should upload this as a separate file labeled 'Revised Manuscript with Track Changes'.An unmarked version of your revised paper without tracked changes. You should upload this as a separate file labeled 'Manuscript'.

We look forward to receiving your revised manuscript.

Kind regards,

A Kakkar

Academic Editor

PLOS ONE

Journal Requirements:

Additional Editor Comments:

This paper evaluates the understanding and attitudes of prescribers in public hospitals in Ghana towards the rational use of medicines, a crucial aspect in mitigating treatment failures, reducing healthcare costs, and minimising drug toxicities and interactions.

Comments:

1. It will be useful to describe in the methods section as to who all are considered as prescribers in Ghana. There is a category designated as 'others' also.

2. A copy of questionnaire used in this study is needed. The authors mention that each correctly answered question was assigned one point. The questionnaire will help the readers in understanding what is meant by 'good knowledge' and positive perception'. What is meant by correct answers???

3. Similarly it is mentioned that "participants who achieved a score of 70% or higher were classified as having a good level of rational prescribing knowledge

or perception, respectively." since there were only 5 questions for perception (text mentions six(Q26-30)????) how was 70% determined. The methodology regarding rounding off etc needs to be mentioned. Also were only 5 questions considered enough for gauging perception of prescribers?

4. There is a term 'reference source available' used. What is meant by this? This needs to be described in text.

5. Do all healthcare facilities included in this study have their own STGs and drug bulletins? Any previous studies on availability of these in Ghana can be cited in the text.

6.p-value mentioned in line 227 is not in alignment with the values in table. The positive perception mentioned is not significant.

7. Interestingly, with respect to frequency of drug bulletin updates, odds of good knowledge and positive perception are lower in participants where there is an annual update as compared to when they are never updated!

8. Discussion: statement "This finding contradicts a systematic review which indicates that medications prescribed by nurse prescribers are as effective and lead to positive patient outcomes, similar to physician prescribing" is difficult to correlate with findings of this study that looks at prescriber knowledge and attitudes.

9. Similarly: "This finding aligns with a study conducted in Turkey, which revealed that nurses with eleven or more years of experience were less prone to medication errors" seems to link perception with medication errors.

10. Limitations of the study such as cross sectional design, limited generalisability, self reported data, social desirability bias etc. need to be included in the discussion. Acknowledging these limitations is important for contextualising study's findings within broader landscape of research on RUM.

Reviewers' comments:

Reviewer's Responses to Questions

**Comments to the Author**

1. Is the manuscript technically sound, and do the data support the conclusions?

Reviewer #1: Partly

Reviewer #2: Yes

2. Has the statistical analysis been performed appropriately and rigorously? 

Reviewer #1: Yes

Reviewer #2: Yes

3. Have the authors made all data underlying the findings in their manuscript fully available?

Reviewer #1: Yes

Reviewer #2: No

4. Is the manuscript presented in an intelligible fashion and written in standard English?

Reviewer #1: Yes

Reviewer #2: Yes

5. Review Comments to the Author

Reviewer #1: The study addresses an important issue of rational use of medicines by assessing the knowledge and perception of the prescribers of a region of Ghana. The study design and analysis seem appropriate. However since the questionnaire is not accessible, it is difficult to assess the appropriateness of the questions to test knowledge and perception. Moreover the details about who were involved in preparation of the tool and what were the sources referred to prepare the same is not mentioned and also who validated the questionnaire is not known.

Reviewer #2: It is a well written manuscript on and such research is relevant to highlight importance of rationally use of medicines. Please ensure data is available through a supplementary file, I could not see it attached as supplementary material.

6. PLOS authors have the option to publish the peer review history of their article (what does this mean?). If published, this will include your full peer review and any attached files.

Reviewer #1: No

Reviewer #2: **Yes: **Ratinder Jhaj

---

## [Author Response · Author response to Decision Letter 0]

30 Apr 2024

We appreciate the opportunity to respond to the comments provided by the academic editor and reviewers regarding our manuscript titled "Assessment of Knowledge and Perception of Prescribers towards Rational Medicine Use in the Ashanti Region of Ghana." We have carefully considered each point raised and have made revisions accordingly. Below, we provide point-by-point responses to the concerns raised:

1. We have duly noted the diverse range of healthcare professionals involved in the medication prescribing process in Ghana, as outlined in the methods section on page 6. We acknowledge the inclusion of mental health nurses, disease control officers, and community health nurses categorized as “Others” in the data analysis.

2. A copy of the study questionnaire, including the bolded correct answers to the knowledge questions, has been uploaded as per the reviewer's suggestion. 

3. The threshold score for determining good knowledge and perception has been clarified to be 70%, with corrections made in the text regarding the number of items used to assess perception and knowledge. Additionally, it has been indicated in the methods section that knowledge scores, when converted to percentages, have been rounded to one decimal place using standard rounding rules. The perception score analysis has been redone using ordinal logistic regression. This has also been indicated in the method section.

4. We have changed the term "reference source available" to “reference source accessible” under the results subsection. This clarification aims to facilitate better understanding and ease of referencing when necessary.

5. The existence of a national Standard Treatment Guideline (STG) developed by the Ministry of Health of Ghana, along with the process of formulary list development by individual hospitals, has been appropriately addressed. We acknowledge the limitation of not verifying respondents' answers regarding the availability of STG in their hospitals and this has been noted in the study limitations statement.

6. The statement regarding the association between functional Drug and Therapeutic Committees (DTCs) and RUM knowledge has been corrected as per the reviewer's suggestion.

7. The discussion section has been revised to address the unexpected finding regarding the relationship between updates to the drug bulletin and prescriber knowledge. We acknowledge the need for further research to understand the underlying factors contributing to this discrepancy.

8. We have revised the discussion to provide context from prior research on knowledge levels among Ghanaian prescribers, particularly comparing doctors and Community Health Officers, as well as findings from similar studies conducted in Pakistan.

9. The discussion has been redone after the ordinal regression analysis aligned with our study's results.

10. A study limitation statement has been included before the conclusion, highlighting the cross-sectional design's limitations and potential biases introduced by self-reported data.

We believe these revisions have strengthened the manuscript and addressed the concerns raised by the academic editor and reviewers. We thank the editorial team for their valuable feedback and consideration of our submission.

---

## [Decision Letter · Decision Letter 1]

19 Jul 2024

PONE-D-23-33862R1Assessment of knowledge and perception of prescribers towards rational medicine use in the Ashanti Region of GhanaPLOS ONE

Dear Dr. Djochie,

Thank you for submitting your manuscript to PLOS ONE. After careful consideration, we feel that it has merit but does not fully meet PLOS ONE’s publication criteria as it currently stands. Therefore, we invite you to submit a revised version of the manuscript that addresses the points raised during the review process.

Kindly see the comments from reviewer 2 

"In Discussion: In the first sentence it is better to avoid adjectives like 'solid' for comprehension.

Moreover, knowledge and not comprehension was assessed, so the term 'knowldge' needs to be used consistently.

Similarly in Conclusion the term 'understanding' may be replaced with 'knowledge' " 

As this is a very minor revision request, kindly submit the correction as soon as possible to allow prompt processing of the article for publication.

We look forward to receiving your revised manuscript.

Kind regards,

Obed Kwabena Offe Amponsah, PharmD, Ph.D.

Academic Editor

PLOS ONE

Journal Requirements:

Reviewers' comments:

Reviewer's Responses to Questions

**Comments to the Author**

1. If the authors have adequately addressed your comments raised in a previous round of review and you feel that this manuscript is now acceptable for publication, you may indicate that here to bypass the “Comments to the Author” section, enter your conflict of interest statement in the “Confidential to Editor” section, and submit your "Accept" recommendation.

Reviewer #1: All comments have been addressed

Reviewer #2: All comments have been addressed

2. Is the manuscript technically sound, and do the data support the conclusions?

Reviewer #1: Yes

Reviewer #2: Yes

3. Has the statistical analysis been performed appropriately and rigorously? 

Reviewer #1: Yes

Reviewer #2: Yes

4. Have the authors made all data underlying the findings in their manuscript fully available?

Reviewer #1: Yes

Reviewer #2: Yes

5. Is the manuscript presented in an intelligible fashion and written in standard English?

Reviewer #1: Yes

Reviewer #2: Yes

6. Review Comments to the Author

Reviewer #1: The authors have addressed all the queries point wise and the explanations provided are satisfactory

Reviewer #2: Almost all reviewer comments have been addressed. Only one samll correction in Discussion and Conclusion.

In Discussion: In the first sentence it is better to avoid adjectives like 'solid' for comprehension.

Moreover, knowledge and not comprehension was assessed, so the term 'knowldge' needs to be used consistently.

Similarly in Conlusion the term'understanding may be replaced with 'knowledge'

7. PLOS authors have the option to publish the peer review history of their article (what does this mean?). If published, this will include your full peer review and any attached files.

Reviewer #1: No

Reviewer #2: **Yes: **Ratinder Jhaj

---

## [Author Response · Author response to Decision Letter 1]

20 Jul 2024

1. In the Discussion section, the first sentence has been rewritten to remove the phrase “solid comprehension” and now reads: "Overall, most participants demonstrated adequate knowledge and a positive perception of the concept of RUM."

2. The term "knowledge" has been used consistently throughout the discussion, replacing "comprehension" and other similar words to avoid ambiguity.

3. In the Conclusion, the term “understanding” has been replaced with “knowledge” as suggested by the reviewer.

4. The references list has been checked to ensure it is complete.

---

## [Editor Report · Decision Letter 2]

24 Jul 2024

Assessment of knowledge and perception of prescribers towards rational medicine use in the Ashanti Region of Ghana

PONE-D-23-33862R2

Dear Dr. Djochie,

We’re pleased to inform you that your manuscript has been judged scientifically suitable for publication and will be formally accepted for publication once it meets all outstanding technical requirements.

Kind regards,

Obed Kwabena Offe Amponsah, PharmD, Ph.D.

Academic Editor

PLOS ONE

Additional Editor Comments (optional):

Thank you for your prompt action on the prior revision request and all the best in future manuscripts.
---

## [Editor Report · Acceptance letter]

29 Jul 2024

PONE-D-23-33862R2 

PLOS ONE

Dear Dr. Djochie, 

I'm pleased to inform you that your manuscript has been deemed suitable for publication in PLOS ONE. Congratulations! Your manuscript is now being handed over to our production team.

Kind regards, 

on behalf of

Dr. Obed Kwabena Offe Amponsah 

Academic Editor

PLOS ONE